# Corneal Edema after Cataract Surgery

**DOI:** 10.3390/jcm12216751

**Published:** 2023-10-25

**Authors:** Celeste Briceno-Lopez, Neus Burguera-Giménez, M. Carmen García-Domene, M. Amparo Díez-Ajenjo, Cristina Peris-Martínez, M. José Luque

**Affiliations:** 1Department of Optics and Optometry and Vision Sciences, Faculty of Physics, Universitat de València, Dr. Moliner 50, E-46100 Burjassot, Spain; neus.burguera@uv.es (N.B.-G.); m.carmen.garcia-domene@uv.es (M.C.G.-D.); amparo.diez@uv.es (M.A.D.-A.); maria.j.luque@uv.es (M.J.L.); 2Cátedra Alcon—FOM—UVEG, Universitat de València, Dr. Moliner 50, E-46100 Burjassot, Spain; 3Anterior Segment and Cornea and External Eye Diseases Unit, Fundación de Oftalmología Médica, Av. Pío Baroja 12, E-46015 Valencia, Spain; cristinaperismartinez0@gmail.com; 4Surgery Department, Faculty of Medicine, Universitat de València, Av. Blasco Ibáñez 15, E-46010 Valencia, Spain

**Keywords:** corneal edema, cataract surgery, visual performance, pachymetry, optical coherence tomography, refractive changes

## Abstract

This systematic review investigates the prevalence and underlying causes of corneal edema following cataract surgery employing manual phacoemulsification. A comprehensive search encompassing databases such as PubMed, Embase, ProQuest, Cochrane Library, and Scopus was conducted, focusing on variables encompassing cataract surgery and corneal edema. Two independent reviewers systematically extracted pertinent data from 103 articles, consisting of 62 theoretical studies and 41 clinical trials. These studies delved into various aspects related to corneal edema after cataract surgery, including endothelial cell loss, pachymetry measurements, visual performance, surgical techniques, supplies, medications, and assessments of endothelial and epithelial barriers. This review, encompassing an extensive analysis of 3060 records, revealed significant correlations between corneal edema and endothelial cell loss during phacoemulsification surgery. Factors such as patient age, cataract grade, and mechanical stress were identified as contributors to endothelial cell loss. Furthermore, pachymetry and optical coherence tomography emerged as valuable diagnostic tools for assessing corneal edema. In conclusion, this systematic review underscores the link between corneal edema and endothelial cell loss in manual phacoemulsification cataract surgery. It highlights the relevance of factors like patient demographics and diagnostic modalities. However, further research is essential to unravel the complexities of refractive changes and the underlying mechanisms.

## 1. Introduction

Corneal edema was previously defined as an abnormal swelling of the cornea’s main layers: the epithelium, stroma, and endothelium [1,2]. However, due to technological advances, this swelling process has now been reduced to one definition: corneal edema is an inflammation of the stroma caused by excessive hydration that affects light transmission above 90%, refraction, transparency, and visual performance [3,4,5,6,7,8,9]. These patients have decreased best-corrected visual acuity (BCVA) and frequency-selective sensitivity loss (CSF) [3,10].

Corneal hydration is influenced by five main factors: tear evaporation, the barrier function of the epithelium and endothelium, stromal swelling pressure, the endothelial pump, and intraocular pressure [10,11,12,13]. The cornea is bounded on the outside by the epithelium and on the inside by the endothelium, which are in contact with the tear film and aqueous humor, respectively. Both tissues play a role as a barrier and keep the stroma hydrated. The stroma is composed of collagen fibrils encased in regularly spaced glycosaminoglycans and mucopolysaccharides that tend to absorb fluid and swell [1,5,14,15,16,17,18,19,20]. Epithelial and endothelial cells are intimately involved in regulating the circulation of fluids and electrolytes in the stroma [1,5,9,11,14,17,21,22]. Therefore, in order to avoid excess hydration by the tear film or aqueous humor, the epithelial and endothelial layers must act as blockages that can regulate water exchange in the stroma so that it is properly hydrated [5,14,22,23,24].

The most common cause of corneal edema is phacoemulsification with intraocular lens implantation, which happens to be the most commonly performed surgery in the world for treating cataracts [7,25,26,27,28]. Corneal edema can occur due to endothelial damage during surgery [5,7,15,29,30]. Endothelial decompensation during cataract surgery has been reported in the literature [20,24,26,30,31,32,33]. However, some of the causes of corneal swelling are classified according to factors directly related to the patient (low endothelial cell density, cataract grade, shallow anterior chamber depth), to the surgical procedure (nucleus extraction method, effective power time of ultrasound for lens extraction, type of viscoelastic used, vitreous loss and rupture of the posterior capsule) [12,26,27,30,32,33,34], or to intraocular lens implantation (chronic iritis, secondary glaucoma, peripheral anterior synechiae, intraocular lens subluxation) [31,32,35].

Finally, it is important to further investigate this corneal disease, as chronic or prolonged swelling processes may lead to bullous keratopathy (BK) or pseudophakic cystoid macular edema (PCME) [15,20,33,35,36,37,38,39]. BK is an incurable disease primarily caused by endothelial decompensation, typically after surgery, mechanical stress, glaucoma, or diabetes mellitus, making keratoplasty the only treatment option to restore the patient’s vision [15,31,35,36,37,40]. The aim of this analysis is to investigate and understand the cause and prevalence of the occurrence of corneal edema after manual phacoemulsification in cataract surgery.

## 2. Materials and Methods

### 2.1. Search Databases

The search databases used were PubMed, Embase, ProQuest, Cochrane Library, and Scopus. The Embase and Medline databases were searched together via the Embase website. Relevant publications were also carefully reviewed. The search was performed using the keywords cornea, corneal, edema, oedema, swelling, treatment, surgery, phacoemulsification, and causes. In addition, all available reviews on the treatment of corneal edema were manually reviewed for additional relevant studies. Articles published up to 31 December 2022 were considered. No restrictions or limitations were placed on the search.

The terms used in the search strategy were progressively limited to the keywords shown in Table 1.

### 2.2. Inclusion and Exclusion Criteria

Studies were considered relevant if they described the edema process, its characterization, medical treatment, or physiological corneal changes in the presence of corneal edema. These pairwise meta-analyses followed the Preferred Reporting Items for Systematic Reviews and Meta-Analyses (PRISMA) method. Studies that were potentially relevant according to the eligibility criteria were selected. The inclusion criteria were (a) experimental studies on the prevalence and characterization of corneal edema after cataract surgery, (b) analytical observational studies (control vs. experimental, cohort) or descriptive studies (transversal), (c) inclusion of comparison groups, and (d) articles in English, French, Spanish, Catalan, Portuguese, or Italian. The exclusion criteria were (a) studies with single clinical cases, case series, reviews, or informative articles; (b) studies that did not specifically investigate the association between corneal edema and phacoemulsification; (c) correspondence letters; (d) systematic and literature reviews; and (e) duplicate studies (repeated in the search databases).

Two independent reviewers (CBL and NBG) reviewed the titles and abstracts of the retrieved articles in the pre-selection phase of the meta-analysis. The articles selected for the preparation of this systematic review were read independently by CBL and NBG. Disagreements regarding the inclusion of studies were resolved through discussion. If disagreements persisted, third and fourth reviewers (MCGD and MADA) were consulted.

### 2.3. Data Extraction

Quality assessment and data extraction of the selected studies were performed by two independent reviewers, CBL and NBG. Data were integrated by CBL using a standardized data extraction form that focused on the main subject, with no secondary ocular diseases associated, to ensure feasibility and reliability. The second stage of data extraction involved the acquirement of relevant data from the included studies in the first stage. Thus, study characteristics such as study design, sample size, population characteristics, and intervention details were extracted. In addition, outcome data such as primary and secondary outcomes, measures used to assess outcomes, and study results were also extracted. Discrepancies between data extraction results were reviewed by CBL and NBG and resolved through discussion. Similarly, third and fourth reviewers were consulted when agreement could not be reached between the first two reviewers.

## 3. Results

A total of 3060 records were initially identified through a systematic search of electronic databases. After using automated tools to remove duplicates and ineligible articles, 3056 articles were screened based on their titles and abstracts. A total of 103 articles met the eligibility criteria and were considered for full-text review. Finally, 41 clinical trials and 62 theoretical articles on corneal oedema/edema and/or phacoemulsification were deemed eligible and included in the systematic review. The PRISMA flow diagram (Figure 1) provides an overview of the study selection process. The included studies were conducted in different countries and published over a range of years. Study designs varied, and sample sizes ranged from small pilot studies to large randomized controlled trials. We assessed the quality of the included studies using the Cochrane Risk-of-Bias Tool (see Table 2) and the Newcastle–Ottawa Scale. In the case of the Cochrane Risk-of-Bias Tool, the theoretical articles will not appear since this tool is only for clinical trials.

However, 2939 studies found during the search were excluded due to the presence of other corneal pathologies associated with the occurrence of corneal edema, such as keratoplasty (DMEK or DSAEK) and Fuch’s Endothelial Corneal Dystrophy (FECD). In line with this exclusion, case reports were also excluded. The systematic review allowed us to include only articles related to uneventful phacoemulsification and the development of corneal edema to answer our main hypothesis. Thus, the inclusion criteria limited the sample to healthy patients aged 50 years and older with cataracts in one or both eyes.

## 4. Discussion

### 4.1. Endothelial Cell Loss (ECL)

Corneal edema is associated with endothelial cell loss during phacoemulsification, which is caused by endothelial pump or barrier dysfunction. Endothelial cells contain specialized Na+/K+-ATPase pumps that use oxygen and energy to keep the cornea clear and transparent. These pumps move fluid out of the stroma in order to maintain the water content under 78%, while the endothelial cells’ tight junctions form a barrier that prevents fluid from flowing into the cornea. Because endothelial cells do not replicate, after injury, adjacent cells enlarge and spread to cover the affected area, resulting in a lower cell density (ECD) and hexagonality (HEX) but a higher coefficient of variation (CV) [39,64]. Based on the literature, a minimum of 500 cells/mm^2^ is required to maintain corneal transparency [35,39]. Specular microscopy is used to measure these endothelial cell parameters, but it is only effective on the central 1 mm^2^ of the cornea, and it cannot be used to assess all of the corneal layers in order to determine the degree of corneal edema [13,41,42,64]. Intraoperative mechanical stress can be caused by stress and thus trauma to the corneal endothelium. Traumatic edema is when swelling occurs after surgery due to excessive stress on the corneal tissue or for accidental reasons [5,6,12,17,21,33,37,65]. This stress can be produced by an excess of power in the phacoemulsifier that damages endothelial cells. Prolonged surgical times can also cause stress, for example, due to intraoperative difficulties in extracting a dense cataractous lens. Other factors leading to excessive tissue stress include narrow anterior chamber depths, which increase the risk of contact with the corneal endothelium, as well as accidental reasons, like poor patient cooperation or movement. In addition, the resolution of edema depends on the severity of the trauma. Penetrating damage may be irreversible, as it is caused by the loss of endothelial cells, which impairs the pumping function of this layer and therefore causes stromal swelling or bullous keratopathy in severe cases [18,23,31,37,38,39].

In surgeries such as phacoemulsification, the most common intraocular surgery worldwide, mechanical stress can be caused by the technique used, the materials used (size and shape), the density of the nucleus, the intraoperative medications, and the surgeon’s experience [5,7,10,17,27,30,37,38,39,43,44,65,67]. Viscoelastic substances are routinely used in cataract surgeries and anterior segment procedures to avoid complications since they have a protective effect on the corneal endothelium [12,41,44,66,88]. There are two types of viscoelastic substances, cohesive and dispersive, with the latter reported to be more effective in preventing endothelial damage during cataract surgery [41,88]. Because viscoelastic substances are nontoxic, nonpyrogenic, and noninflammatory and have the same osmolality as the cornea or aqueous fluid (305 and 300 mOsmol/kg, respectively), they should not interfere with normal intraocular tissue metabolism or intraocular pressure (IOP). Contact with a hypo-osmotic solution causes a breakdown of intercellular junctions and intracellular edema, resulting in excessive imbibition of fluids into the stroma and corneal edema [39,41,68]. However, corneal edema may also be due to a disrupted epithelial barrier. Epithelial edema can occur if excessive local anesthetics or ultrasound vibration is used during phacoemulsification. The integrity of both the epithelial and endothelial layers is critical to keeping corneal deturgescence and a balanced stroma hydration [1,10,21,23,30,45,68,69].

On the other hand, there are factors that contribute to ECL, such as age, nucleus firmness, and the grade of the cataract [10,12,24,30,44,46,67,69,70,89]. Choi and Han found that preoperative lens nucleus hardness and postoperative corneal edema were the most significant predictive factors for endothelial cell loss at 10 years [70]. Eyes with a higher cataract grade showed an annual endothelial cell loss of 3% or more. While age alone did not show an independent association, it is likely linked to the cataract grade in elderly patients. Mechanical stress was also reflected in postoperative corneal edema, which was associated with accelerated long-term endothelial loss. In addition, they attempted to measure endothelial cell loss in patients who had undergone cataract surgery using a mathematical model:ECL_10years_ = 5.435 + 5.606 × PCEG_OCTET_ + 6.425 × NF,

This model predicts edema with a 10-year time lapse (ECL: endothelial cell loss (in percentage); PCEG: postoperative corneal edema grade using OCTET classification; NF: nuclear firmness based on the Emery–Little system).

This mathematical model, developed by Choi and Han, allows for predicting long-term endothelial cell loss after cataract surgery based on clinically observable factors like postoperative edema and nucleus hardness [70]. The authors found that their model accounted for 39.7% of the variation in endothelial cell loss at 10 years post-surgery. While promising, this specific quantitative approach does not yet seem to be widely used, requiring further validation across diverse patient populations. However, it represents an initial attempt to quantitatively estimate endothelial damage after cataract surgery based on clinically observable factors. Moreover, based on their model, the authors estimated that endothelial cell density may decrease by anywhere from 164 to 523 cells/mm2 after uneventful cataract surgery, depending on crystalline lens firmness. They also claimed that phacoemulsification causes nearly three times the endothelial cell loss that occurs in natural aging, with the loss rising to 2.06 ± 1.36% annually in operated eyes compared to 0.6% in non-operated eyes. Other studies estimated that the long-term endothelial cell loss following cataract surgery may be 0.09 to 2.5% greater than physiological loss related to aging alone, potentially persisting for at least 10 years [6,44,47,64,67,70].

### 4.2. Pachymetry

Pachymetry has become the most common means of detecting and diagnosing corneal edema based on central corneal thickness (CCT) [2,21,27,43,44,65,69,75,76,77,89], along with subjective examination with the slit lamp, suggesting that an increase above 10% of the original pachymetry is an indicator of edema [1,27,48,65,77,90,91]. Hence, for an average CCT of 550 microns, an increase of 10%, which is 605 microns, after cataract surgery would be indicative of corneal edema. Some studies suggest that this parameter may overlook subclinical or mild corneal swelling [74,75,78,92]. Yet, due to the strong correlation between CCT and endothelial cell loss following cataract surgery that has been documented in the literature, pachymetry has emerged as the gold standard when corneal edema occurs [19,38,39,42,69].

Recent study evidence, on the other hand, has shown that optical coherence tomography (OCT) is another tool for objectively measuring corneal edema [9,13,37,38,47,78]. Moreover, since its development, OCT has become an indispensable tool in clinical practice, especially Swept-Source OCT, due to its depth capacity in capturing images and the speed required by the device to acquire high-resolution images without contact with the patient. In addition, OCT is able to objectively measure ocular structures and capture architectural features, such as the width, length, thickness, alignment, and gaps of the corneal incision, which are particularly important in phacoemulsification with manual incision [13,38]. OCT also allows ophthalmologists to objectively assess the evolution of the corneal incision or corneal thickness after surgery, among other intraoperative factors, as they have shown a strong correlation with edema onset [13,38]. In addition, these authors attempted to classify edema by stromal opacity using the densitometry unit of the device and showed a strong correlation between optical density, pachymetry, and BCVA, but only for FECD [78]. In contrast, Zéboulon et al. found a good correlation between OCT and Pentacam as an indicator of edema in normal corneas using pachymetry maps, but these two devices may perform poorly in thinner or thicker corneas or in subclinical or mild corneal swelling [92]. Another study by Suzuki and colleagues suggests using the corneal volume (CV) measured by Pentacam to assess corneal endothelial cell damage after phacoemulsification because endothelial cell density appears to represent only a small portion of the injury inflicted on this layer and is insufficient to properly represent changes in the entire cornea [42].

In line with these statements, other authors point out that the corneal thickness limit is 650 µm because the device’s algorithms use the refractive index to calculate corneal thickness, and the measurements would otherwise be unreliable [47]. This limit is also disadvantageous for severe corneal edema with opacities measured with ultrasound pachymetry or Pentacam [3].

### 4.3. Visual Performance

Ishikawa et al. [74] suggested, in contrast to other studies, that the presence of corneal edema does not affect visual acuity, although visual symptoms may still be present [1,10,20,37,49,66,74,91,93,94]. In addition, Díez-Ajenjo et al. demonstrated that despite the absence of changes in corneal radii, refractive changes occur that are not known, and the underlying mechanisms are unclear [3]. In addition, the authors suggest that further studies are needed to determine whether the refractive index of the cornea changes due to increased hydration during the swelling process. They found that even in corneal edema, the corneal tissues are affected to varying degrees. Furthermore, there is a direct correlation between the refractive index and stromal fluid uptake when the cornea is swollen, suggesting that corneal edema cannot be explained by changes in corneal thickness only. On the other hand, Meek et al. found that the refractive index can change due to hydration, and therefore, it is logical to assume that the corneal refractive index shifts due to corneal swelling according to Gladstone and Dale’s law [95]. Refractive indices are particularly important in optical measurements such as pachymetry or automatic refraction because these devices assume a refractive index that does not necessarily reflect the true refractive index, especially in the case of swollen corneas [3,95].

Some authors have reported that the quality of vision improves over time as the transparency of the cornea increases as the swelling process recedes [1,7,10,11,20,22,23,39,79]. Corneal edema thus leads to an overall reduction in spatial frequencies, resulting in a significant loss of sensitivity acuity under mesopic conditions, which could be related to the loss of transparency and an increase in light scattering in patients with corneal swelling [3,10,23,80,94,96]. However, De Juan et al. [81] pointed out that significant refractive changes occur in the first week after cataract surgery and stabilize in the following weeks. They also revealed that corneal edema causes a hyperopic shift due to corneal swelling, although this effect diminishes within the first two weeks and changes to myopia as the edema subsides, ideally approaching emmetropia.

### 4.4. Mechanical Trauma

Corneal edema is the most common side effect of ocular surgery, with a prevalence, according to some authors, from 6.2% to 11.3% [36,37], most commonly after phacoemulsification or keratoplasty in severe FECD [6,66,82,97]. Preoperative risk factors for phacoemulsification include a low endothelial cell density, which should always be greater than 1000 cells/mm^2^, and corneal thickness, which should be less than 640 µm [5,20,35,42,47,82]. Precautions must be taken during phacoemulsification to avoid corneal edema caused by trauma phacoemulsifiers, lens hardness, short anterior chamber depths, retained viscoelastic fragments of the lens, irrigating solutions and their temperature, phacoemulsification time and energy, IOL insertion, and the surgeon’s expertise, which is especially important in patients with pseudoexfoliation or previous endothelial damage (guttae) [6,9,12,17,34,44,45,47,50,65,98,99].

Traumatic edema is regarded as edema that develops following surgery, in this case, cataract surgery, as a result of excessive strain on the tissue or for other incidental reasons [5,6,17,30,45,51,52,79,96,98]. The severity of the trauma determines the resolution of the edema. Therefore, the loss of endothelial cells can result in penetrating damage that is irreversible, impairing the pump function of this layer and, in severe cases, causing stromal swelling or bullous keratopathy [20,31,34,86,96]. The most common intraocular surgery in the world, phacoemulsification, can cause mechanical stress depending on the technique used, the material used (size and shape), the use of intraoperative drugs, and the experience of the surgeon [5,7,12,17,33,45,52,53,54,67,86].

The phacoemulsification technique, as was already mentioned, is an important point in the development of corneal edema. According to some authors, microincision cataract surgery (MICS) results in a smaller corneal incision, which has been shown to reduce wound misalignment and to carefully preserve the corneal integrity after the cataract removal procedure [13,46,53,55,56,99,100]. They reported fewer changes in corneal endothelial cell loss, pachymetric parameters, and corneal edema over the short term, as well as fewer induced corneal aberrations in the long term [13,55,56,82]. Other authors, on the other hand, argue that there is no difference in these parameters between MICS and conventional phacoemulsification [56,67].

Furthermore, it has been reported in the literature that the method of cataract surgery, as well as the nucleus fracture and extraction techniques, have an effect on the onset of postoperative corneal edema. Recent research indicates that phacoemulsification and extracapsular cataract surgery (ECCE), more specifically manual small-incision cataract surgery (SICS), are both safe and effective procedures for restoring optimal visual performance in cataract patients. The primary goal of phacoemulsification extraction techniques (including pre-chop, stop-and-chop, divide-and-conquer, drill-and-crack, and soft-shell) is to tear apart the nucleus using mechanical force with the energy released by the needle as it buries itself inside the lens and to aspirate the cortex debris out of the eye. Some of these techniques have demonstrated various benefits in terms of ultrasound power (USP) and effective phacoemulsification time (EPT) in order to cause less endothelial cell loss. As a result, these methods have been modified for the treatment of dense cataracts, which remain a challenge in the cataract surgery process [24,28,32,35,38,49,54,100]. Zhao et al. investigated this specifically, comparing reverse-chopper nucleus extraction in dense cataracts to conventional stop-and-chop under the same conditions. The reverse chopper required less EPT and USP, resulting in no ultrasonic energy released and thus less endothelial cell damage with a lower prevalence of corneal edema following cataract surgery [28].

Epithelial edema may develop if topical anesthetics or ultrasonic vibration power are used excessively during phacoemulsification [29,33,66]. Long-lasting surgery may cause Descemet’s membrane to detach, which will result in stromal edema [7,83,84,101]. A low endothelial density or shallow anterior chambers have also been noted as risk factors for postoperative edema development. In every situation, a thorough preoperative examination is required. Additionally, abrupt changes in IOP may cause an increase in stromal volume as a result of excessive water intake [17,41]. This hyperhydration causes a hypertonic state that directly affects the endothelial pump’s ability to function, which could account for glaucoma patients’ propensity to experience corneal swelling following phacoemulsification [11].

### 4.5. Treatment

The treatment will vary according to the etiology, aiming to eliminate the underlying pathology, if possible. The standard treatment for anterior segment inflammation and, as a result, corneal edema caused by inflammation or infection is topical corticosteroids [5,51,53,57,58,59,60,61,62,89,95,102].

During the inflammation process, lymphangiogenesis occurs, which is the inhibition of lymphatic endothelial cell proliferation. Glucocorticoids inhibit corneal lymphangiogenesis by suppressing the onset of pro-inflammatory cytokines and macrophage infiltration, as well as the proliferation of lymphatic endothelial cells [9,57,58,60]. Furthermore, some research suggests that corticosteroids may activate the endothelial pump function, thereby preventing corneal edema [53,59,60]. Corticosteroids, such as Dexamethasone, should be used with caution in patients with high IOP values and low anterior chamber penetration because they have been shown to raise IOP by 10 mmHg or more in these patients [53,58,61,63]. Despite this, recent research indicates that Loteprednol, a modern corticosteroid, is a safe option due to its low impact on IOP, though it is contraindicated in viral, fungal, or mycobacterial infections. Increased and decreased IOP alters the pressure in the stroma, leading to increased imbibition, corneal swelling, and increased corneal thickness, all of which may affect the accuracy of Goldmann Applanation Tonometry (GAT) measurements [71,85,103]. Sriram and Tai discovered that increased IOP causes fluids to accumulate intercellularly within the cornea, resulting in epithelial edema [58,103].

Hypotony, on the other hand, only causes stromal swelling; although the mechanism is unknown, it is thought to be caused by a tautness loss in the tissue. Furthermore, the ocular tissues’ relaxation could result in folds in Descemet’s and Bowman’s layers [83,103]. The flow and composition of the fluids are altered in chronic hypotony, causing disruptions in corneal metabolism, a lack of nutrient transport in the tissues, hypoxia, dysfunction of the blood–aqueous barrier, and eventually, endothelial pump dysfunction [17,45,66,103]. Since a carbonic anhydrase inhibitor lowers pressure, it is prescribed when there is a high IOP because it prevents corneal edema from developing by lowering IOP. Inhibiting the carbonic anhydrase pump may have a direct effect on the stroma by decreasing outflow from this layer to the anterior chamber, resulting in stromal hyperhydration, particularly in eyes with compromised endothelial function [5,12,39]. Carbonic anhydrase, along with the Na+/K+-ATPase pump, yields a pH buffer through which a lactate osmotic gradient drives water out of the stroma and into the aqueous humor [5,12,18,21,37,86,91]. When hypotony causes edema, it is usually a consequence of an over-filtering trabeculectomy and is less likely to be chronic. The first case is treatable by resuturing the scleral flap or compressing the conjunctival autograft once more [74]. The purpose of hypertonic solutions, also known as hyperosmolar solutions, is to draw fluids out of the cornea, causing dehydration via the osmotic gradient and increasing tear film tonicity in epithelial corneal edema and restoring transparency [18,20,41,50,93,96]. To do so, the epithelium must be regular and functioning properly, as it serves as a semipermeable membrane that allows only water to pass while keeping the electrolytes in the cornea. As a result, the solutes do not penetrate the epithelium completely or only partially, attracting diffusible water from the bullae and preventing stromal hyperhydration.

Crosslinking (CXL) may be used as an alternative to keratoplasty in severe cases of corneal edema, such as postoperative BK (PBK) and FECD, according to recent research [34,51,76,87]. CXL is a photopolymerization technique that uses a photosensitizing substance like riboflavin and UV light to induce additional crosslinks within the matrix of collagen fibers, increasing corneal stiffness. Some studies suggest that CXL may compact the corneal stroma and reduce swelling in edematous corneas by improving endothelial pump function or reducing corneal hydration [51,72,76,87]. For example, Laborante et al. [72] treated six patients with visually impairing corneal edema using CXL alone or with amniotic membrane transplantation. They reported improved corneal transparency and visual acuity in all patients, concluding that CXL could delay the need for keratoplasty. However, the effects of CXL on corneal edema are not consistent across studies. Coskunseven et al. [73] found no significant improvement in corneal transparency in seven patients with PBK treated with CXL. Additional randomized controlled trials are warranted to further evaluate the efficacy of CXL for managing corneal edema before it can be recommended as a standard therapy.

Additionally, therapeutic contact lenses are typically used to relieve discomfort or improve vision when there are epithelial irregularities or keratopathies, such as bullous keratopathy [5,6,72]. Nowadays, either rigid or silicone hydrogel soft contact lenses are hypoxia-safe, and the prescription varies depending on the case. Despite their comfort, rigid contact lenses increase oxygen transmissibility and smooth the corneal surface from higher irregularities. It is sometimes necessary to prescribe therapeutic contact lenses in conjunction with antibiotics to prevent contact-lens-induced infections, especially when soft contact lenses are prescribed. Contact lenses should be worn carefully to avoid lens-induced hypoxia, which can result in excess lactate and protons (stroma), as well as inflammatory eicosanoids (epithelium), according to some studies [11,18,71,72]. Both protons and lactate function as osmotic stimulants and change the pH of the stroma directly, which decreases endothelial pump activity and starts the process of corneal swelling [11,18,22,68,73,91].

In addition to these assertions, due to the importance that oxygen has in achieving normal corneal metabolic function and visual performance, current research has shown that postoperative eye patching may cause corneal edema and a slower pace of visual recovery due to oxygen shortage after successful cataract surgery [11,39].

## 5. Conclusions

This thorough review of the literature offers a comprehensive exploration of the realm of corneal edema. Notably, endothelial cells take center stage in maintaining the delicate balance of corneal transparency and hydration. As confocal microscopy becomes a clinical standard, endothelial cell density emerges as a predictive factor for post-surgical corneal swelling. In contrast, changes in pachymetry, alongside subjective observations through biomicroscopy, stand as significant objective markers for corneal edema progression.

Despite the wealth of knowledge accumulated, a puzzling aspect persists. Corneal edema significantly impacts visual performance by affecting aspects such as visual acuity and contrast sensitivity. However, a notable uncertainty looms: What precisely is the impact of corneal swelling on an individual’s vision? The permanence or transience of these changes remains a question mark.

Intriguingly, the prevailing classification of edema remains largely subjective, raising a pertinent concern, particularly in the context of subclinical corneal swelling. This subjective nature underscores the challenge of achieving a consensus, especially in instances where subtlety is paramount.

As we navigate this complex landscape, these questions guide us toward a deeper understanding of corneal edema’s implications and the quest for standardized assessment methods.

## 6. Study Limitations

This systematic review avoided time constraints by including a wide range of studies up to December 2022. On the other hand, bias may have occurred in the language of the selected articles, although this systematic review included languages other than English. However, the degree of heterogeneity among the included studies was high, so a possible publication bias could not be excluded. Finally, a publication bias could also be present, as the studies included in this work were found using filters such as “pair-reviewed”. In addition, positive or statistically significant results are more likely to be published than studies with negative or non-significant results. These limitations highlight the need for further research in this area.

## Figures and Tables

**Figure 1 jcm-12-06751-f001:**
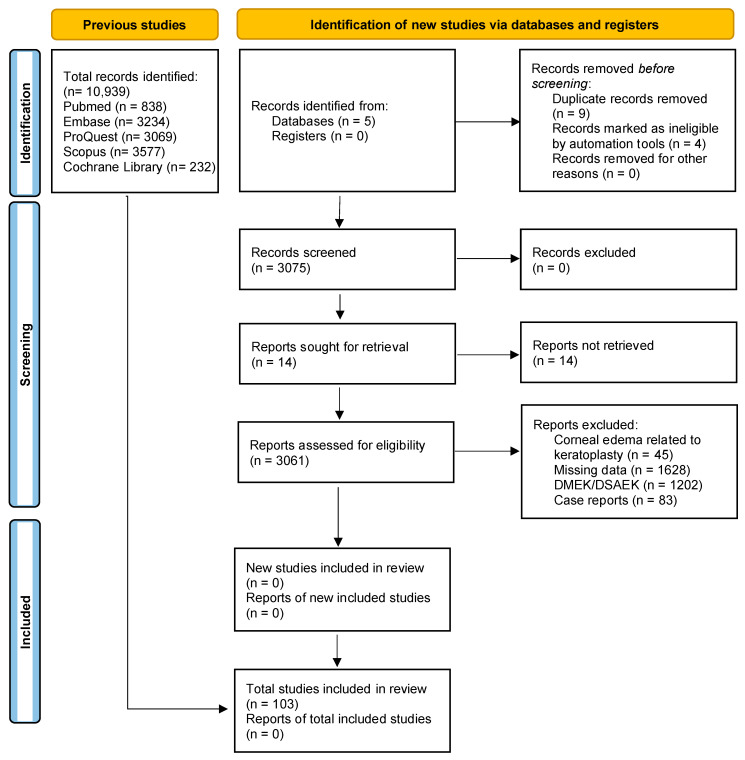
Study flow diagram. DMEK: Descemet’s Membrane Endothelial Keratoplasty. DSAEK: Descemet’s Stripping Automated Endothelial Keratoplasty.

**Table 1 jcm-12-06751-t001:** Search strategy and keywords. CCT: Central corneal thickness. ECD: Endothelial cell density.

Corneal Edema		After		Cataract Surgery		Variables Measured
“Corneal edema” OR “corneal oedema” OR “edematic cornea” OR “edematous cornea” OR “corneal edema incidence” OR “corneal inflammation” OR “corneal swelling”	**AND**	“after” OR “consequence” OR “following” OR “from” OR “post” OR “pursuant” OR “result” OR “subsequent” OR “succeeding” OR “successive”	**AND**	“Cataract surgery” OR “cataract extraction” OR “phacoemulsification” OR “manual phacoemulsification” OR “aphakia” OR “pseudophakic” OR “manual-incisioned phacoemulsification”	**AND**	“Central corneal thickness” OR “CCT” OR “corneal thickness” OR “pachometry” OR “pachymetry” OR “Endothelial cell density” OR “ECD” OR “endothelial cell damage” OR “endothelial cell loss” OR “corneal endothelium” OR “endothelial cells” OR “specular microscopy”

**Table 2 jcm-12-06751-t002:** Table based on the Cochrane Risk-of-Bias Tool to review the studies’ quality assessment. Studies that were not applicable to a given category are not shown in that category but are included in other applicable categories.

**RANDOM SEQUENCE GENERATION** **Selection Bias (Biased Allocation to Interventions) Due to Inadequate Generation of a Randomized Sequence.**		
References	Judgment	Support for judgment
[7,11,13,18,19,20,21,26,28,29,30,32,35,37,39,41,42,43,44,45,46,47,48,49,50,51,52,53,54,55,56,57,58,59,60,61,62,63]	Low risk	Referring to a random number table; minimization
[3,8,10,12,25,27,31,36,38,40,64,65,66]	High risk	Non-random component in the sequence generation process
[67,68,69,70]	Unclear	Insufficient information about the sequence generation process to permit judgment of “Low risk” or “High risk”.
**ALLOCATION CONCEALMENT** **Selection bias (biased allocation to interventions) due to inadequate concealment of allocations prior to assignment.**		
References	Judgment	Support for judgment
[7,11,13,18,19,20,21,26,30,35,37,39,41,42,43,44,45,46,47,49,52,56,57,58,59,60,61,62,63]	Low risk	Central allocation (including telephone, web-based, and pharmacy-controlled randomization); sequentially numbered drug containers of identical appearance; sequentially numbered, opaque, sealed envelopes.
[3,8,10,12,25,27,31,32,36,38,40,50,53,54,55,64,65,66]	High risk	Using an open random allocation schedule (e.g., a list of random numbers); assignment envelopes were used without appropriate safeguards (e.g., if envelopes were unsealed or nonopaque or not sequentially numbered); alternation or rotation; date of birth; case record number; any other explicitly unconcealed procedure.
[28,29,67,68,69,70]	Unclear	Insufficient information to permit judgment of “Low risk” or “High risk”. This is usually the case if the method of concealment is not described or not described in sufficient detail to allow a definite judgment—for example, if the use of assignment envelopes is described, but it remains unclear whether envelopes were sequentially numbered, opaque, and sealed.
**BLINDING OF PARTICIPANTS AND PERSONNEL** **Performance bias due to knowledge of the allocated interventions by participants and personnel during the study.**		
References	Judgment	Support for judgment
[20,37,44,45,47,48,58,60,61,62,71,72,73]	Low risk	No blinding or incomplete blinding, but the review authors judge that the outcome is not likely to be influenced by lack of blinding; blinding of participants and key study personnel ensured, and unlikely that the blinding could have been broken.
[3,7,8,10,11,12,13,18,19,21,25,27,29,30,31,32,35,36,38,39,40,41,42,43,46,49,50,51,52,53,54,55,56,57,59,63,64,65,66,67,69,70,74]	High risk	No blinding or incomplete blinding, and the outcome is likely to be influenced by lack of blinding; blinding of key study participants and personnel attempted, but likely that the blinding could have been broken, and the outcome is likely to be influenced by lack of blinding.
[26,28,68]	Unclear	Insufficient information to permit judgment of “Low risk” or “High risk”; the study did not address this outcome.
**BLINDING OF OUTCOME ASSESSMENT** **Detection bias due to knowledge of the allocated interventions by outcome assessors.**		
References	Judgment	Support for judgment
[20,44,45,47,49,51,52,54,56,58,60,61,62,68,71,72,73,74,75,76,77,78,79,80,81,82,83,84,85]	Low risk	No blinding of outcome assessment, but the review authors judge that the outcome measurement is not likely to be influenced by lack of blinding; blinding of outcome assessment ensured, and unlikely that the blinding could have been broken.
[3,7,8,10,11,12,13,18,19,21,25,27,29,30,31,32,35,36,37,38,39,40,41,42,43,46,48,50,53,55,57,59,63,64,65,66,67,69,70,86]	High risk	No blinding of outcome assessment, and the outcome measurement is likely to be influenced by lack of blinding; blinding of outcome assessment, but likely that the blinding could have been broken, and the outcome measurement is likely to be influenced by lack of blinding.
[26,28]	Unclear	Insufficient information to permit judgment of “Low risk” or “High risk”; the study did not address this outcome.
**INCOMPLETE OUTCOME DATA** **Attrition bias due to amount, nature, or handling of incomplete outcome data.**		
References	Judgment	Support for judgment
[3,7,8,10,11,13,18,19,20,21,26,28,30,35,36,37,38,39,40,41,42,43,44,45,46,47,49,50,51,52,53,54,55,56,57,58,59,60,62,63,65,66,67,68,69,70,71,72,73,74,75,76,77,78,79,80,81,82,83,84,85,86,87]	Low risk	No missing outcome data; reasons for missing outcome data unlikely to be related to true outcome (for survival data, censoring unlikely to be introducing bias); missing outcome data balanced in numbers across intervention groups, with similar reasons for missing data across groups; for dichotomous outcome data, the proportion of missing outcomes compared with observed event risk not enough to have a clinically relevant impact on the intervention effect estimate; for continuous outcome data, plausible effect size (difference in means or standardized difference in means) among missing outcomes not enough to have a clinically relevant impact on observed effect size; missing data have been imputed using appropriate methods.
[48,61,64]	High risk	Reason for missing outcome data likely to be related to true outcome, with either imbalance in numbers or reasons for missing data across intervention groups; for dichotomous outcome data, the proportion of missing outcomes compared with observed event risk enough to induce clinically relevant bias in intervention effect estimate; for continuous outcome data, plausible effect size (difference in means or standardized difference in means) among missing outcomes enough to induce clinically relevant bias in observed effect size; “As-treated” analysis performed with substantial departure of the intervention received from that assigned at randomization; potentially inappropriate application of simple imputation.
[12,25,27,29,31,32]	Unclear	Insufficient reporting of attrition/exclusions to permit judgment of “Low risk” or “High risk” (e.g., number randomized not stated, no reasons for missing data provided); the study did not address this outcome.
**SELECTIVE REPORTING** **Reporting bias due to selective outcome reporting.**		
References	Judgment	Support for judgment
[3,7,8,10,11,12,13,18,19,20,21,26,27,28,29,30,32,35,36,37,38,39,40,41,42,43,44,45,46,47,48,49,50,51,52,53,54,55,56,57,58,59,60,61,62,63,64,65,66,67,68,69,70,71,72,73,74,75,76,77,78,79,80,81,82,83,84,85,86,87]	Low risk	The study protocol is available and all of the study’s pre-specified (primary and secondary) outcomes that are of interest in the review have been reported in the pre-specified way; the study protocol is not available, but it is clear that the published reports include all expected outcomes, including those that were pre-specified (convincing text of this nature may be uncommon).
[25]	High risk	Not all of the study’s pre-specified primary outcomes have been reported; one or more primary outcomes are reported using measurements, analysis methods, or subsets of the data (e.g., subscales) that were not pre-specified; one or more reported primary outcomes were not pre-specified (unless clear justification for their reporting is provided, such as an unexpected adverse effect); one or more outcomes of interest in the review are reported incompletely, so they cannot be included in a meta-analysis; the study report fails to include results for a key outcome that would be expected to have been reported for such a study.
[31]	Unclear	Insufficient information to permit judgment of “Low risk” or “High risk”. It is likely that the majority of studies will fall into this category.
**OTHER BIASES** **Bias due to problems not covered elsewhere in the table**.		
References	Judgment	Support for judgment
[3,7,8,10,11,12,13,18,19,20,21,26,27,28,29,30,32,35,36,37,39,40,41,42,43,44,45,46,47,48,49,50,52,53,54,55,56,57,58,59,60,61,62,63,64,65,66,67,68,71,72,73,74,75,76,77,78,79,80,81,82,83,84,85,86,87]	Low risk	The study appears to be free of other sources of bias.
[25,31,38,69,70]	High risk	Had a potential source of bias related to the specific study design used; has been claimed to have been fraudulent; or had some other problem.
	Unclear	Insufficient information to assess whether an important risk of bias exists or insufficient rationale or evidence that an identified problem will introduce bias.

## Data Availability

The datasets produced and/or analyzed in the course of this study are accessible upon a justified request to the corresponding author.

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
