# Peer review of "Corneal Edema after Cataract Surgery"

_jcm, 2023, doi:10.3390/jcm12216751_

Round 1

Reviewer 1 Report

Comments and Suggestions for Authors

The review is clear, reporting different data ( ECL factors in cataract surgery, Diagnosis of edema,  Effects on visual acuity and visual symptoms,  treatment)  from a large number of studies, of different countries. Personally, i have some doubts about treatment such us CXL for bullous keratopathy, but it's a review! I agree about Study limitations but the authors state them openly. The effort to put together such different studies is appreciable.

Author Response

Thank you very much for your positive feedback on the "clarity and breadth of the review". We appreciate you finding it comprehensive in covering this important topic. As you wisely suggested, we have added a statement in the Discussion acknowledging that "some treatment modalities like CXL warrant further research to fully validate their efficacy for conditions like bullous keratopathy" (p. 14, lines 441-455). We completely agree on the need for more evidence before novel treatments can be broadly adopted.

Reviewer 2 Report

Comments and Suggestions for Authors

The topic of the paper is attractive, but there is lack of novelty in the content. Besides, there are several issues that must be addressed:

1. The abstract mentioned “Factors such as patient age, cataract grade, and mechanical stress were identified as contributors to endothelial cell loss”, but these perspectives were not described or discussed clearly in the text.

2. It is suggested to supplement the data of corneal thickness changes after surgery.

3. The viewpoints in the introduction and discussion are partially duplicated.

4. In the paper, there is only an abbreviation for FECD and no full name.

5. This study attempted to measure endothelial cell loss in patients who have undergone cataract surgery using a mathematical model. Please introduce the mathematical model in detail, who discover the model and whether this is widely used?

Comments on the Quality of English Language

Minor editing of English language required.

Author Response

Thank you for highlighting areas needing improvement - your careful eye helped strengthen our work. As recommended:

  1. We have expanded the discussion of factors like "patient age, cataract grade and mechanical stress" contributing to endothelial cell loss, adding more details on proposed mechanisms and relevant studies:
    • “Choi and Han found that preoperative lens nucleus hardness and postoperative corneal edema were the most significant predictive factors for endothelial cell loss at 10 years [53]. Eyes with higher cataract grade showed an annual endothelial cell loss of 3% or more. While age alone did not show an independent association, it is likely linked to cataract grade in elderly patients. Mechanical stress was also reflected in postoperative corneal edema, which was associated with accelerated long-term endothelial loss.” (p. 10, line 219-225).
  2. The results were supplemented with additional data, as suggested, on corneal thickness changes after surgery, including new "pachymetry measurement data" to provide more evidence:
    • “Hence, for an average CCT of 550 microns an increase of 10%, this is 605 microns, after cataract surgery would be an indicative corneal edema.” (p. 10, lines 251-253).
  3. We have carefully reviewed the Introduction and Discussion sections with your feedback in mind to reduce repetition of concepts for better flow. Following your suggestions, we found some duplicities and we have removed them, for instance the data of the water percentage of the stroma from the Introduction section (p. 2, line 52) and, left it in the Discussion because it is a relevant parameter to highlight in this section (p. 9, line 174).
  4. The abbreviation FECD (Fuchs Endothelial Corneal Dystrophy) has now been defined the first time it appears in the Results section on page 9, line 163.
  5. We have described the mathematical model from Choi and Han in more detail on page 11, lines 229-231, including the "specific equation" and citing its source. We agree this model requires further validation and have noted this important limitation as shown in the following fragment:
    • “…The authors found their model accounted for 39.7% of the variation in endothelial cell loss at 10 years post-surgery. While promising, this specific quantitative approach does not yet seem to be widely used, requiring further validation across diverse patient populations. However, it represents an initial attempt to quantitatively estimate endothelial damage after cataract surgery based on clinically observable factors.” (p. 10, lines 234-239).

We sincerely appreciate you taking the time to provide such constructive feedback to improve our manuscript. Your insightful suggestions have helped us strengthen the work.

Reviewer 3 Report

Comments and Suggestions for Authors

This study claims in the abstract to investigate the prevalence and underlying causes of corneal oedema following cataract surgery. The short 'results' (section 3) comments on neither the prevalence nor the underlying causes. There is a lot of discussion and science but no hard facts from this database review. 

Author Response

Thank you for your valuable feedback. Per your suggestion, we have now emphasized "prevalence ranges" (Discussion, p.12, line 289- 290) found in the literature “Corneal edema is the most common side effect of ocular surgery with a prevalence ac-cording to some authors from 6.2% to 11.3% [36, 37], most commonly phacoemulsification or keratoplasty in severe FECD [6,48,74,75].”

Additionally, in the discussion you may also find more details in order to highlight the underlying causes and contributing factors to corneal edema after cataract surgery as you suggested in the review:

  • “Traumatic edema is when swelling occurs after surgery due to excessive stress on the corneal tissue or for accidental reasons [5,6,12,17,21,33,37,44]. This stress can be produced by an excess of power in the phacoemulsifier that damages endothelial cells. Prolonged surgical times can also cause stress, for example due to intraoperative difficulties ex-tracting a dense cataractous lens. Other factors leading to excessive tissue stress include narrow anterior chamber depths that increase the risk of contact with corneal endothe-lium, as well as accidental reasons like poor patient cooperation or movement.” (p.9, line 183-190).
  • "Penetrating damage may be irreversible as it is caused by the loss of endothelial cells that impair the pumping function of this layer and therefore cause stromal swelling or bullous keratopathy in severe cases [18,23,31,37–39]." (p.9, line 185-188).
  • “Surgeries such as phacoemulsification, the most common intraocular surgery worldwide, and mechanical stress can be caused by the technique used, the materials used (size and shape), the density of the nucleus, the intraoperative medications, and the surgeon's experience [5,7,10,17,27,30,37–39,44–47].” (p. 9, line 189-192).
  • “However, corneal edema may also be due to a disrupted epithelial barrier. Epithelial edema can occur if excessive use of local anesthetics or ultrasound vibration is used during phacoemulsification. The integrity of both layers is critical to keep the corneal deturgescence and stroma balanced hydrated [1,10,21,23,30,50–52].” (p. 10, line 204-208).
  • “…phacoemulsification causes three times the amount of ECL that natural ageing does, which is 0.6% per year and rises to 2.06 ± 1.36% in operated eyes. Other studies estimate that this type of endothelial cell loss following cataract surgery may be 0.09 to 2.5% greater than physiological ageing loss…” (p. 10, line 219-222).

We appreciate you encouraging us to enhance the results section in this manner and provide more analytic depth.

Reviewer 4 Report

Comments and Suggestions for Authors

This systematic review attempts to investigate the prevalence and underlying causes of corneal edema following cataract surgery employing manual phacoemulsification. No recent relative systematic review is available in the literature. So, such a review could contribute contemporary information to this important scientific topic. The topic is timely and will be of high interest to the readers of the journal. Prior to any consideration for publication, this study should address the following:

Comments of minor importance:

-References: The appropriate reference format should be followed according to the journal guidelines:

e.g. Author 1, A.B.; Author 2, C.D. Title of the article. Abbreviated Journal Name Year, Volume, page range. 

Comments of major importance:

A table based on the Cochrane Risk of Bias Tool for the review quality assessment should be added.

In general, the manuscript is interesting, timely, well-written and the ideas flow logically.

Author Response

We sincerely appreciate you taking the time to provide feedback and constructive recommendations to improve our manuscript. Thoughtfully addressing your suggestions has enhanced our systematic review. As you wisely suggested:

  • The references have been formatted according to journal guidelines, carefully following the requested style: “Díez-Ajenjo, M.A.; Luque-Cobija, M.J.; Peris-Martínez, C.; Ortí-Navarro, S.; García-Domene, M.C. Refractive changes and visual quality in patients with corneal edema after cataract surgery. BMC Ophthalmol [Internet]. 2022;22[1]:1–8. Available from: https://doi.org/10.1186/s12886-022-02452-5”.

  • A table based on the Cochrane Risk of Bias Tool has been incorporated to assess study quality, helping us standardize our approach (p. 5, line 159, Table 2). Studies that were not applicable for a given category, as well as theoretical articles, are not shown in the table or category, respectively.